# Sensitive Electrochemical Detection of Bioactive Molecules (Hydrogen Peroxide, Glucose, Dopamine) with Perovskites-Based Sensors

**Imane Boubezari [1], Ali Zazoua [1] , Abdelhamid Errachid [2] and Nicole Jaffrezic-Renault [2],***

[1]  Laboratory of Applied Energetics and Materials, University of Jijel, Ouled Aissa 18000, Algeria; boubezari.imen@gmail.com (I.B.); a_zazoua@univ-jijel.dz (A.Z.)

[2]  Institute of Analytical Sciences, University of Lyon, 69100 Villeurbanne, France; abdelhamid.errachid-el-salhi@univ-lyon1.fr

*  Correspondence: nicole.jaffrezic@univ-lyon1.fr

**Abstract:** Perovskite-modified electrodes have received increasing attention in the last decade, due to their electrocatalytic properties to undergo the sensitive and selective detection of bioactive molecules, such as hydrogen peroxide, glucose, and dopamine. In this review paper, different types of perovskites involved for their electrocatalytic properties are described, and the proposed mechanism of detection is presented. The analytical performances obtained for different electroactive molecules are listed and compared with those in terms of the type of perovskite used, its nanostructuration, and its association with other conductive nanomaterials. The analytical performance obtained with perovskites is shown to be better than those of Ni and Co oxide-based electrochemical sensors. Main trends and future challenges for enlarging and improving the use of perovskite-based electrochemical sensors are then discussed.

**Keywords:** perovskite; voltammetry; hydrogen peroxide; glucose; dopamine

## 1. Introduction

For the integration in point-of-care testing (POCT) systems [1], such as commercial kits available for monitoring patients' glucose levels (i.e., Accu-Chek®from Roche Diabetes Care Company) [2], there is an urgent need for precise, sensitive, portable, and cost-effective technologies for the detection of bioactive molecules. In this group of chemical compounds, hydrogen peroxide is a cancer biomarker, because, in comparison to normal cells, cancer cells are characterized by an increased $H_2O_2$ production rate and an impaired redox balance, thereby affecting the antitumoral immune response [3]. Diabetes mellitus is due to an abnormal level of glucose in the blood, and this level should be frequently monitored [4]. Dopamine (DA) is one of the most important catecholamines, present in the human central nervous system. Its depletion that should be monitored leads to neurodegenerative diseases such as Parkinson's disease [5].

Electrochemical sensors are good candidates for their integration in POCT systems [6], due to their easy miniaturization and the low-cost instrumentation that could be interfaced with smart phones. All these bioactive molecules can be sensitively detected by enzymatic electrochemical sensors [7–9]. The main drawback of these enzymatic sensors is the stability of enzymes and their activities being modified by the immobilization procedure, by the pH value, or by the presence of inhibiting agents. The use of non-enzymatic electrochemical sensors should be highly required. During the last decade, perovskite nanomaterials have shown electrocatalytic properties, and bioactive molecules could be easily sensitively and selectively detected using perovskite-based electrodes, which present the advantage of stability of the sensors.

Perovskite oxides were discovered by Gustav Rose in the Ural mountains in 1839; the most common formula of perovskites is $ABO_3$, with A being an alkali metal or a lanthanide,

B being a transition metal, and O being the oxygen ion. The charges of A and B ions should be equivalent to the whole charge of the oxygen ions. The tolerance factor should be in the range of 0.8–1.0 with the radii of A and B ions greater than 0.090 nm and 0.051 nm, respectively. The cubic structure of the perovskite is stabilized by the 6-fold coordination of the B cation and the 12-fold coordination of the A cation (Figure 1).

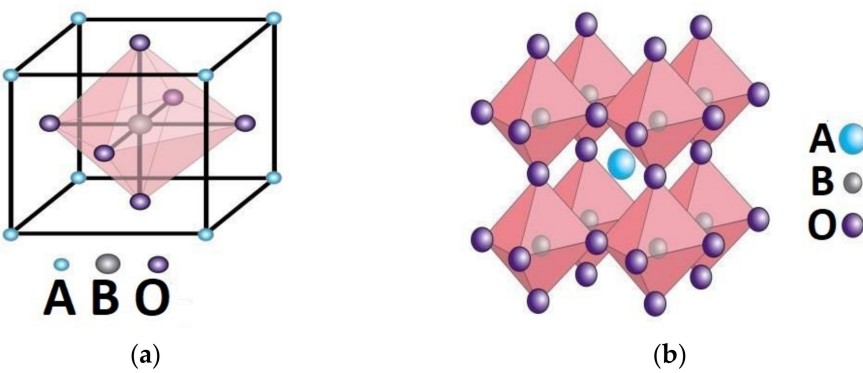

(a)　　　　　　　　　　　　　　　　　　　　　　　　　　　(b)

**Figure 1.** (**a**) Cubic mesh of perovskite; (**b**) three-dimensional (3D) stacking in the ideal cubic structure.

Some distortions may occur in the ideal cubic structure, leading to orthorhombic, tetragonal, rhombohedral, or hexagonal structures.

Perovskite oxides present a diversity of electrical properties from insulating to semi-conducting metallic and superconducting properties. They also present magnetic and optical properties [10]. A lot of devices are then conceived from perovskite oxides: phot-chromic, electro-chromic, image storage, switching, filtering, and surface acoustic wave signal-processing devices. In addition, they have been used as catalysts in different applications such as engine-exhaust gas treatment and hydrogen evolution reaction.

Several recent review papers were devoted to the sensory applications of perovskite based on their electrical and of optical properties [11–17]. This review paper is based on an exhaustive list of perovskite-based electrochemical sensors for the enzyme-free detection of hydrogen peroxide, glucose, and DA. In each case, the involved mechanism is described, and the analytical performance of the obtained sensor is presented. The perovskite formula leading to the lowest detection limit is highlighted, as well as the one leading to the most selective detection. The improvement brought by the association with other nanomaterials is also shown. The analytical performance of the perovskite-based sensors and those of nickel oxide and cobalt oxide-based sensors are compared. The main trends and future challenges are discussed.

## 2. Methods for the Synthesis and Characterization of Perovskites

### 2.1. Sol-Gel Synthesis

The sol-gel synthesis of precursors is one of the procedures used for the synthesis of perovskites [18]. It is based on the Pechini method that involves two chemical reactions: with nitrate salts of the metal ions being mixed according to the stoichiometry, the complexation of metal ions occurs with the addition of citric acid, and the polyesterification of the complexes is obtained with the addition of ethylene glycol [19]. A viscous solution is obtained after heating to 130 °C in an argon flow under stirring for about 10 h. The temperature is then raised to 150–200 °C to obtain a foam-dried mass which has to be ground in a mortar. This precursor is calcined at different temperatures of 600, 850, and 1000 °C in a muffle furnace, depending on the final targeted product.

### 2.2. Microwave Irradiation Process

The microwave irradiation process shows many advantages such as rapid reaction velocity, uniform heating, cleanness, and high energy efficiency. The conditions used for microwave preparation are 2.45 GHz, with a maximum output power of not less than

1 kW. Dielectric materials can absorb the microwave energy and transform it into heat energy directly through polarization and dielectric loss in the interior of the materials. Single-phase manganese-based perovskites are simply obtained from nitric solutions by a denitration process under microwave irradiation [20].

### 2.3. Coprecipitation Process

In the coprecipitation process [18], different types of precursors are employed: oxides, alkoxides, inorganic salts, and nitrates. The processing parameters (pH, coprecipitation rate, washing, drying, and temperature of synthesis) are controlled which results in homogeneous and weakly agglomerated nanopowders. They can be sintered at temperatures as low as 1250 °C and for short times (1–2 h) for the obtaining of perovskites of near the theoretical density.

### 2.4. Solid-State Synthesis Technique

$LaFeO_3$ nanoparticles are prepared through the solid-state synthesis technique using mechanical ball milling [21]. Stoichiometric amounts of $La_2O_3$ and $Fe_2O_3$ metal oxide precursors (molar ratio: 1:1) are transferred to a planetary ball mill. Wet milling is carried out for 20 h (with toluene as the process control agent). Based on thermal analysis, the ball milled powder is calcined at 900 °C for 2 h and then ground into fine powders in an agate mortar and pestle.

### 2.5. Other Synthesis Techniques

$LaNiO_3$ electrospun nanofibers are prepared by mixing metal salts with polyvinylpyrrolidone (PVP) followed by sequential calcinations [22]. Thin films of $La_{0.5}Sr_{0.5}CoO_{3-\delta}$ are obtained by the pulsed laser deposition (PLD) technique [23].

### 2.6. Characterization Methods

The different phases of the prepared perovskites can be differentiated using X-ray powder diffraction (XRD). In addition, the structure of perovskite can be characterized using single-crystal XRD analysis. Thermal analysis techniques such as thermogravimetry (TGA), differential thermal analysis (DTA), and differential scanning calorimetry (DSC) can be used to test the thermal stability of the prepared perovskites. The different morphological characteristics of the prepared perovskites can be studied using scanning electron microscopy (SEM) and transmission electron microscopy (TEM). In addition, the surface area measurement of the prepared perovskites can be carried out using surface area analysis (BET). Raman spectroscopy allows the determination of vibration modes in relation with molecular dynamics. In addition, the surface chemical groups of the prepared perovskites can be identified using Fourier-transform infrared spectroscopy (FTIR) and X-ray photoelectron spectroscopy (XPS). The frequency-dependent conductivity spectra are determined by using impedance spectroscopy.

## 3. Perovskite-Based Electrochemical Sensors for the Detection of Hydrogen Peroxide

Table 1 shows a summary of Perovskite-based electrochemical sensors for the detection of hydrogen peroxide.

Table 1. Perovskite-based electrochemical sensors for the detection of hydrogen peroxide.

| Type of Perovskite Electrode | Perovskite Preparation | Sensitivity | Linear Range | Detection Limit (LOD) | Lifetime | Applications | References |
|---|---|---|---|---|---|---|---|
| $Sr_{0.85}Ce_{0.15}FeO_3$ Perovskite + Nafion®/SPE | citrate-nitrate smoldering autocombustion | 60 µA/mM/cm² | 0–500 µM | 10 µM | 12 months | | [24] |
| $SmCoO_3$ Perovskite + conductive carbon + Nafion®/GCE | EDTA-citrate complexing sol-gel + calcination | 715 µA/mM/cm² | 0.1–5000 µM | 0.004 µM | | | [25] |
| $La_{0.6}Ca_{0.4}MnO_3$ CPE | Malic acid—nitrate Sol-gel method + calcination | | 0–0.5 mM | | | | [26] |
| $La_{0.7}Sr_{0.3}Mn_{0.75}Co_{0.2}5O3$ CPE | Metal salts mixed with polyvinylpyrrolidone (PVP) Electrospinning and calcination | 1371 µA/mM | 0.5–1000 µM | 0.17 µM | 30 days | Toothpaste Medical hydrogen peroxide | [27] |
| $LaNi_{0.6}Co_{0.4}O_3$ CPE | Citrate-nitrate Sol-gel method + calcination | 1812 µA/mM/cm² | 10 nM–100 µM | 1 nM | 20 days | Toothpaste | [28] |
| $Co_{0.4}Fe_{0.6}LaO_3$ CPE | Citrate-nitrate Sol-gel method | 2376.7 nA/µM | 0.01–800 µM | 2 nM | 3 weeks | | [29] |
| $La_{0.66}Sr_{0.33}MnO_3$ CPE | Microwave irradiation of nitric solution | 1770 µA/M | | | | Cleaning product | [20] |
| $LaNiO_3$ CPE | Metal salts mixed with PVP Electrospun nanofibers | 1135.88 µA/mM/cm² | 0.05–1000 µM | 33.9 nM | 4 weeks | | [22] |
| $LaNi_{0.5}Ti_{0.5}O_3/CoFe_2O_3$ GCE | Citrate-nitrate Sol-gel method + calcination | 3.21 µA/mM/cm² | 0.1 µM–8.2 mM | 23 nM | 4 weeks | Toothpaste | [30] |
| $La_{0.6}Sr_{0.4}Co_{0.2}Fe_{0.8}O_{3-\delta}$ | Citrate-nitrate Sol-gel method + calcination | 580 µA/mM/cm² | 0 mM | 5 µM | | | [31] |
| $La_{0.5}Sr_{0.5}CoO_{3-\delta}$ EMOSFET | Pulsed laser deposition technique | | | 1 mM | | | [23] |
| $La_{0.6}Sr_{0.4}CoO_{3-\delta}$ Perovskite + Nafion®/GCE | EDTA-citrate complexing sol-gel + calcination | 280 µA/mM/cm² | 0.4–3350 µM | 0.12 µM | | | [32] |
| $La_{0.6}Sr_{0.4}CoO_{3-\delta}$ Perovskite + RGO + Nafion®/GCE | EDTA-citrate complexing sol-gel + calcination | 500 µA/mM/cm² | 0.2–3350 µM | 0.05 µM | | | [32] |
| $LaMnO_3$/conductive carbon black GCE | Precipitation method + calcination + carbon coating | 897.6 µA/mM/cm² | 5–5550 µM | 0.807 nM | 30 days | | [33] |
| $Nafion-LaNiO_3$ GCE | Citrate-nitrate Sol-gel method + calcination | | 0.2 µM–50 µM | 35 nM | | Serum samples | [34] |

The first study about the electrochemical detection of hydrogen peroxide using a perovskite was published in 1996 by Shimizu [26]. Among $ABO_3$ perovskites used for the electrochemical detection of hydrogen peroxide (Table 1), the more commonly used are lanthanum-based perovskites, certain being substituted with alkaline earth ions such as calcium and strontium. The frequency of the use of B ions for the $H_2O_2$ detection, among all the published papers (Table 1), is as follows: Co > Ni = Mn > Fe > Ti. A-site ion is mainly La that can be substituted by alkaline ions such as strontium [20,23,27,31,32] or calcium [26].

Figure 2 illustrates the overall steps of the $H_2O_2$ electro-oxidation on the perovskite surface, which involves diffusion, adsorption/desorption, and electro-oxidation reaction.

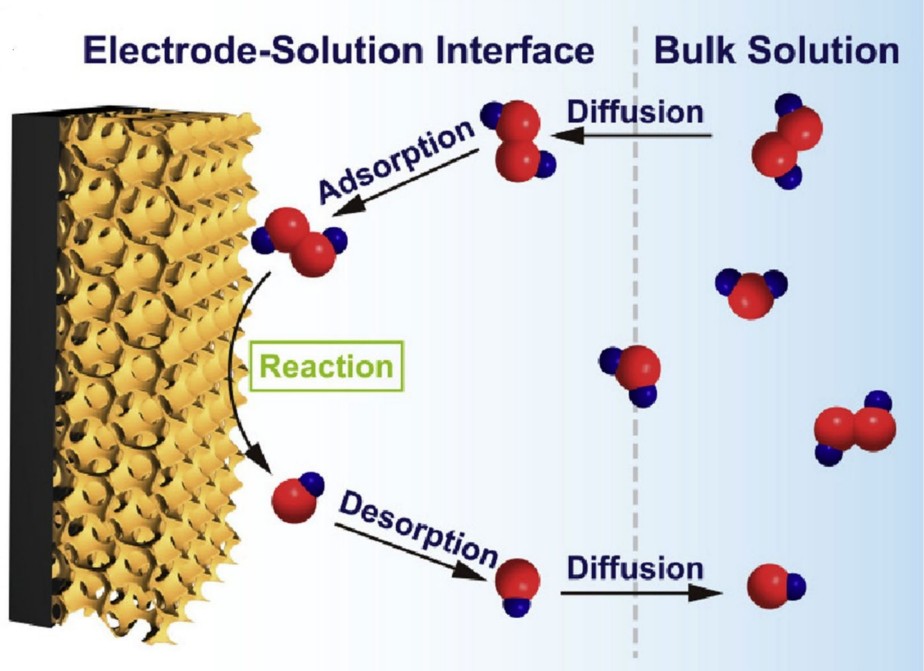

**Figure 2.** Schematic diagram of the overall process steps on a perovskite/glassy carbon electrode (GCE) [21]. Reproduced with the permission from Elsevier.

The complex mechanisms involved in the electrocatalytic oxidation of hydrogen peroxide were deeply analyzed [32]. The partial substitution of the A-site cations by divalent cations such as $Ca^{2+}$, $Sr^{2+}$, and $Ba^{2+}$ can lead to an oxidation of the B-site cation as $B^{3+}$ [24,25,27,32]. The increasing of this substitution also leads to the reduction in oxygen vacancy formation energy which is consistent with the existence of highly oxidative oxygen species [32]. The most possible mechanism of the oxidation of $H_2O_2$ on $La_{0.6}Sr_{0.4}CoO_{3-\delta}$ in 0.1 M NaOH at a potential of 0.3V/Ag/AgCl is presented in Figure 3. Two parallel pathways are involved: $Co^{3+}/Co^{4+}$ redox couple (Figure 3A) and oxygen vacancies formation which allows the transfer of lattice oxygen to the adsorbed intermediates, generating ion superoxide (defined as the lattice-oxygen-mediated oxygen evolution reaction (LOM-OER); Figure 3B).

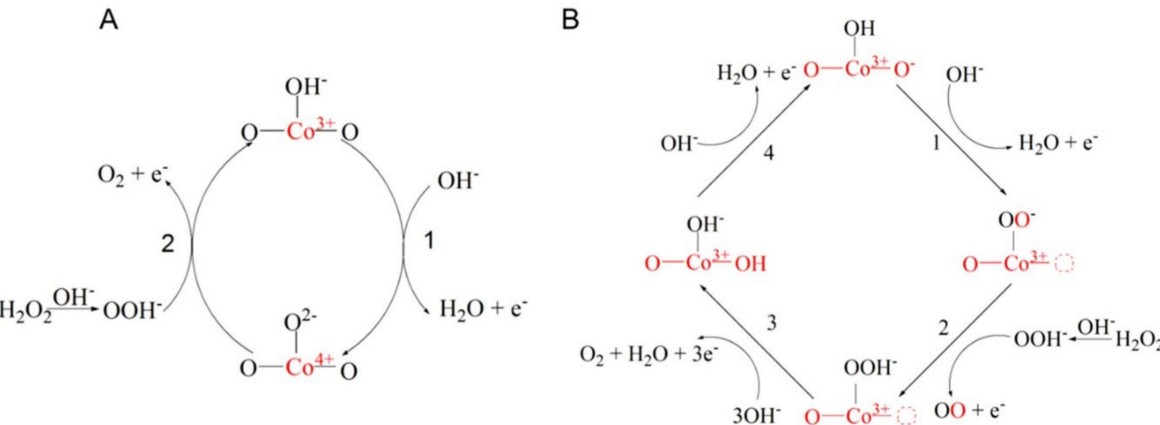

**Figure 3.** Electro-oxidation mechanisms of $H_2O_2$ on $La_{0.6}Sr_{0.4}CoO_{3-\delta}$ that occur simultaneously via the $Co^{3+}/Co^{4+}$ redox couple (**A**) and lattice-oxygen-mediated oxygen evolution reaction (LOM-OER) (**B**) involving oxygen vacancies and superoxide ion ($O_2^{2-}/O^-$) [32]. Reproduced with the permission from Elsevier.

Regarding the analytical performance of the perovskite-based electrochemical sensors for $H_2O_2$, the obtained lower detection limits are 1 nM and 2 nM, obtained with $LaNi_{0.6}Co_{0.4}O_3$ and with $LaCo_{0.4}Fe_{0.6}O_3$, respectively [28,29].

The specific surface area of the perovskite can be increased by different preparation procedures, showing that this parameter is also of importance for the analytical performance. Electrospun nanofibers are prepared by mixing $La_{0.7}Sr_{0.3}Mn_{0.75}Co_{0.25}O_3$ [27] and with $LaNiO_3$ [22] in PVP. Higher sensitivities of detection (more than 1000 $\mu A/mM/cm^2$) and large dynamic ranges until 1000 $\mu M$ are obtained. A three-dimensional (3D) ordered microporous $SmCoO_3$ perovskite is prepared using a poly(methylmethacrylate) colloidal crystal template route [25]. A detection limit of 4 nM and a dynamic range from 0.1 to 5000 $\mu M$ are obtained.

The association of perovskites with conductive nanomaterials allows a decrease of the detection limit. When perovskite $LaMnO_3$ is intimately mixed with a conductive carbon black, forming a composite, a very low detection limit of 0.805 nM is obtained. [33]. It was also noticed that the mixture of $La_{0.6}Sr_{0.4}CoO_{3-\delta}$ with reduced graphene oxide [32] improves the sensitivity by a factor of 2 and decreases the detection limit also by a factor of 2.

The storage stability of these perovskite-based sensors is in the range of one month.

## 4. Perovskite-Based Electrochemical Sensors for the Detection of Glucose

As for hydrogen peroxide detection, numerous works on the detection of glucose (Table 2) are carried out using La-based perovskites. In some of them, La is substituted by alkaline earth such as strontium [31,32].

**Table 2.** Perovskite-based electrochemical sensors for the detection of glucose.

| Type of Perovskite Electrode | Perovskite Preparation | Sensitivity | Linear Range | LOD | Lifetime | Applications | References |
|---|---|---|---|---|---|---|---|
| $Pr_{1.92}Ba_{0.08}NiO_{0.95}Zn_{0.05}O_{4+\delta}$ gold electrode | Citrate-nitrate Sol-gel method + calcination | 101 μA/logC 604 μA/logC | 1.5–50 μM 0.05–7 mM | 0.5 μM | | Human serum | [35] |
| $Sr_{1.7}Ca_{0.3}PdO_3$ Graphite electrode | Glycine-nitrate method + calcination | 306.9 μA/mM 54.17 μA/mM | 5 μM–1.4 mM 1.8–5.6 mM | 8.45 nM | 50 cycles | Human urine samples | [36] |
| $NdNiO_3$ GCE | Hydrothermal method Co-precipitation | 1105.1 μA/mM/cm² | 0.0005–4.6 mM | 0.3 μM | 15 days | Human blood samples | [37] |
| $Sr_2Pd_{0.7}Au_{0.3}O_3$ Graphite electrode | Glycine-nitrate method + calcination | 1.44 x 10⁴ μA/mM/cm² 1639 μA/mM/cm² | 0.4–10 μM 20 μM–100 μM | 2.11 nM 18.5 nM | | Urine samples | [38] |
| $La_{0.6}Sr_{0.4}Co_{0.2}Fe_{0.8}O_{3-\delta}$ | Citrate-nitrate Sol-gel method + calcination | 285 μA/mM/cm² | 0–200 μM | 7 μM | | | [31] |
| $La_{0.6}Sr_{0.4}CoO_{3-\delta}$ Perovskite + Nafion®/GCE | EDTA-citrate complexing sol-gel + calcination | 275 μA/mM/cm² | 5–1500 μM | 0.15 μM | | | [32] |
| La $NiO_3$ CPE | Metal salts mixed with PVP Electrospun nanofibers | 42.321 μA/mM/cm² | 1–1000 μM | 0.32 μM | 4 weeks | | [22] |
| $Co_{0.4}Fe_{0.6}LaO_3$ CPE | Citrate-nitrate Sol-gel method | 1013.8 μA/mM/cm² | 0.05–5 and 5–500 μM | 0.01 μM | 3 weeks | | [29] |
| $LaNi_{0.6}Co_{0.4}O_3$ CPE | Sol-gel method + calcination | 643 μA/mM/cm² | 0.05–200 μM | 8.0 nM | 20 days | | [28] |
| $LaNi_{0.5}Ti_{0.5}O_3$ CPE | Citrate-nitrate Sol-gel method + calcination | 1630 μA/mM/cm² | 0.2–20 μM 0.02–1 mM | 0.07 μM | 40 days | Human blood serum | [39] |
| $La_{0.88}Sr_{0.12}MnO_3$ CPE | Metal salts mixed with PVP Electrospun nanofibers | 1111.11 μA/mM/cm² | 0.05–100 μM | 31.2 nM | 15 days | Serum samples | [40] |
| $La_{0.6}Sr_{0.4}CoO_{3-\delta}$ Perovskite + RGO + Nafion®/GCE | EDTA-citrate complexing sol-gel + calcination | 330 μA/mM/cm² | 2–3350 μM | 0.063 μM | | | [32] |
| $LaTiO_3–Ag_{0.1}$ GCE ECL sensor | Glycine-nitrate method + calcination | 782 μA/mM/cm² | 0.1 μM–0.1 mM | 2.5 nM | 4 weeks | Human serum samples | [41] |
| La $TiO_3–Ag_{0.2}$ GCE | Sol-gel method Butyl titanate in nitric acid + calcination | 784.14 μA/mM/cm² | 2.5 μM–4 mM | 0.21 μM | One month | Blood serum samples | [42] |
| $SrPdO_3$ + AuNP modified graphite electrode | Citrate-nitrate method + calcination | 422.3 μA/mM/cm² | 0.1–6 mM | 10.1 μM | 7 weeks | | [43] |

The complex mechanism for the electrooxidation of glucose on $La_{0.6}Sr_{0.4}CoO_{3-\delta}$ at 0.06 V/Ag/AgCl, in 0.1 M NaOH is presented in [32] and is similar to the proposed mechanism for the electrooxidation of $H_2O_2$, according to two pathways via the $Co^{3+}/Co^{4+}$ redox couple and via the LOM-OER.

The lower detection limit obtained with La-based perovskite (8 nM) is obtained with $LaNi_{0.6}Co_{0.4}O_3$ [28]. The perovskite gives also a lower detection limit for $H_2O_2$ (Table 1), showing the higher catalytic effect of B-site ions Ni and Co. Two La-based perovskites are associated with conductive nanomaterials such as AgNPs [41,42], leading to a very low detection limit (2.5 nM) with an Electrochemiluminescence (ECL) sensor [41].

Other lanthanide A-site ions are used, such as praseodynme [31] and neodyme [33], substituted by alkaline earth such as Ba [35]. Another used A-site ion is an alkaline earth ion, strontium [36,38,43]. The lower detection limit of 2.11 nM was obtained with $Sr_2Pd_{0.7}Au_{0.3}O_3$ [38]. $SrPdO_3$ is also associated with conductive nanomaterials such as AuNPs [43], which allows the selective detection of glucose in the presence of other bioactive interfering species ascorbic acid (AA), uric acid (UA), *N*-acetyl-para-aminophenol or paracetamol (APAP), and DA to be shown through linear sweep voltammetry (LSV) in the case of $SrPdO_3$/AuNPs (Figure 4) [43].

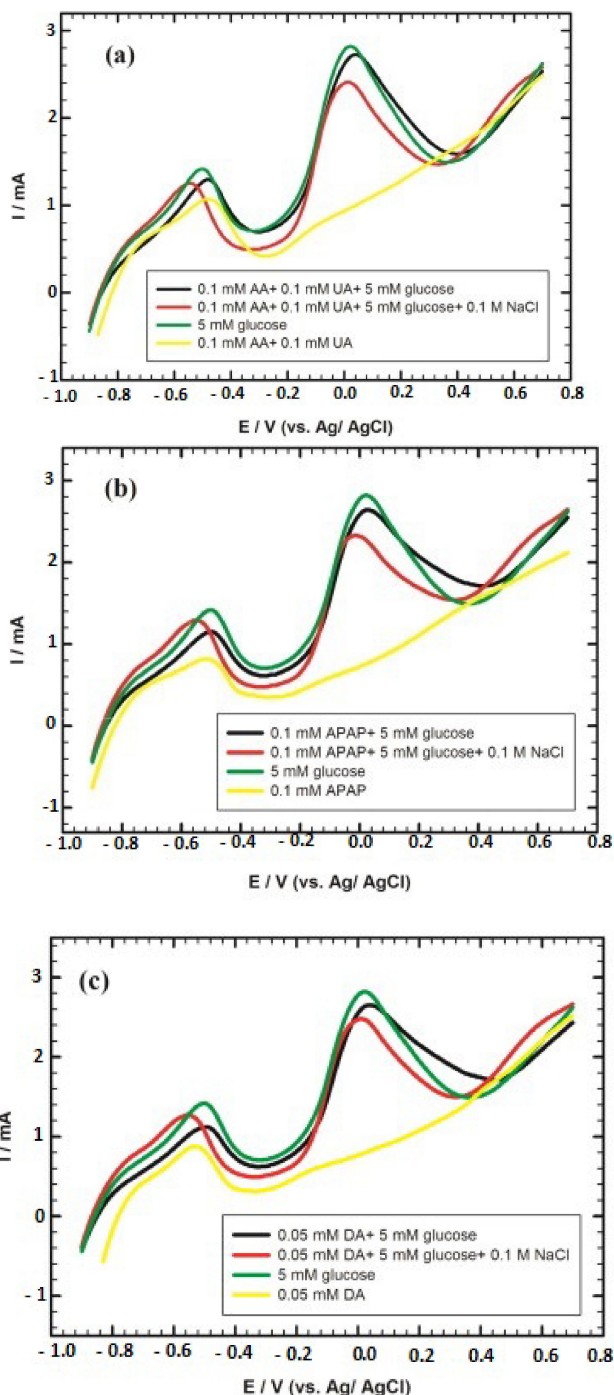

**Figure 4.** (**a**) Linear sweep voltammetry (LSV) curves of 5 mM glucose/0.1 M NaOH in the absence and the presence of 0.1 mM ascorbic acid (AA), 0.1 mM uric acid (UA) and 0.1 M NaCl at graphite/$SrPdO_3$/AuNPs. (**b**) LSV curves of 5 mM/0.1 M NaOH in the absence and the presence of 0.1 mM *N*-acetyl-para-aminophenol or paracetamol (APAP) and 0.1 M NaCl at graphite/$SrPdO_3$/AuNPs. (**c**) LSV curves of 5 mM glucose/0.1 M NaOH in the absence and the presence of 0.05 mM dopamine (DA) and 0.1 M NaCl at graphite/$SrPdO_3$/AuNPs [43]. Reproduced with the permission from Elsevier.

## 5. Perovskite-Based Electrochemical Sensors for the Detection of DA

Seventeen papers from 2014 to 2021 are devoted to the electrocatalytic detection of DA using perovskite nanomaterials (Table 3).

**Table 3.** Perovskite-based electrochemical sensors for the detection of DA and other bioactive molecules.

| Type of Detected Molecule | Type of Perovskite Electrode Perovskite Preparation | Sensitivity | Dynamic Range | LOD | Lifetime | Applications | References |
|---|---|---|---|---|---|---|---|
| DA in the presence of ascorbic acid and uric acid | LaFeO$_3$ GCE Solid state synthesis | | 10–100 μM 120–180 μM | 10 nM | 10 days | Blood samples | [21] |
| DA | LaFeO$_3$ microspheres GCE Sol-gel method Nitrate+ ethylene glycol | | 0.02–1.6 μM | 59 nM | | | [44] |
| DA in the presence of AA and UA | LaCoO$_3$ GCE Hydrothermal process | 0.033 μA/μM | 1–100 μM 0.5–5 mM 0.5–5 mM | 3.53 μM | | | [45] |
| DA, acetaminophen, xanthine | LaFe$_{0.2}$Ni$_{0.8}$O$_3$ Carbon ceramic microelectrode Hydrothermal method Co-precipitation | 0.0109 μA/μM 0.008 μA/μM 0.0274 μA/μM | 6.6–131 μM 10–131 μM 3–115 μM | 2.1 μM 3.2 μM 1.3 μM | | | [46] |
| DA | LaFeO$_3$ Graphite powder Combustion technique | | 5–200 μM | 600 nM | 20 days | | [47] |
| DA | LaMnO$_3$ GCE Natural lemon juice– nitrate sol-gel method | | 1–600 μM | 32 nM | | Human urine saliva | [48] |
| DA | LaNiO$_3$ Citrate + glycine method + calcination CPE | 63.59 μA/mM | 80 nM–20 μM | 9 nM | One month | Urine and serum | [49] |
| Simultaneous detection of DA in the presence of ascorbic acid and uric acid | SrPdO$_3$ CPE Citrate + glycine + urea method + calcination | 0.88 μA/μM | 7–70 μM | 9.3 nM | | Human urine samples | [50] |
| DA | NdFeO$_3$ SPCE Metal salts mixed with PVP + calcination | | 0.5–100 μM 150–400 μM | 0.27 μM | | Urine | [51] |
| DA UA | FeTiO$_3$ coprecipitation + calcination | 1.56 μA/μM/cm$^2$ | 1–90 μM 1–150 μM | 1.3 nM 30 nM | One week | Human serum | [52] |
| DA | NaNbO$_3$ GCE Solvothermal method NbCl$_3$ + ethanol Nb$_2$O$_5$ + Na$_2$CO$_3$ | 99 nA/nM/cm$^2$ 77 nA/nM/cm$^2$ 75 nA/nM/cm$^2$ | 10–50 nM 100–500 nM 1–500 μM | 6.8 nM | | Simulated blood samples | [53] |
| DA | SrTiO$_3$/GO GCE Citrate-nitrate method + calcination | 0.0126 μA/μM/cm$^2$ | 0.05–531 μM | 10 nM | | Blood serum Urine | [54] |

**Table 3.** *Cont.*

| Type of Detected Molecule | Type of Perovskite Electrode Perovskite Preparation | Sensitivity | Dynamic Range | LOD | Lifetime | Applications | References |
|---|---|---|---|---|---|---|---|
| DA | $\beta$-NaFeO$_2$ GCE Solid-state reaction assisted synthesis Na$_2$CO$_3$ + Fe$_2$O$_3$ | DA 27.16 $\mu$A/$\mu$M/cm$^2$ | DA 0.010–40 $\mu$M | DA 2.12 nM | | Simulated blood samples | [55] |
| DA | NdFeO$_3$ Nitrates + PVP + calcination | | 0.5–400 $\mu$M | 270 nM | | | [56] |
| DA | ZnSnO$_3$ nano cube One-pot hydrothermal technique GCE | | 10 nM–5 $\mu$M | 2.6 nM | | Blood serum | [57] |
| DA | CoTMPPyP/Sr$_2$Nb$_3$O$_{10}$ GCE Solid-state reaction assisted synthesis | | 0.02–1.62 mM | 7.6 $\mu$M | 30 days | Human urine | [58] |
| DA | CsPbBr$_3$ nanocrystals encapsulated in conductive silica gel Sol-gel method | | 0.01–10 $\mu$M | 3 nM | | | [59] |

The electrooxidation of DA on $FeTiO_3$ is depicted in Figure 5 [52]. The adsorption of DA on the $FeTiO_3$ surface (including the oxygen deficient sites) occurs, due to the electrostatic force of attraction. The oxidation process, with respect to the applied potential, takes place according to two pathways via the $Ti^{3+}/Ti^{4+}$ and $Fe^{2+}/Fe^{3+}$ redox couples and via the LOM-OER.

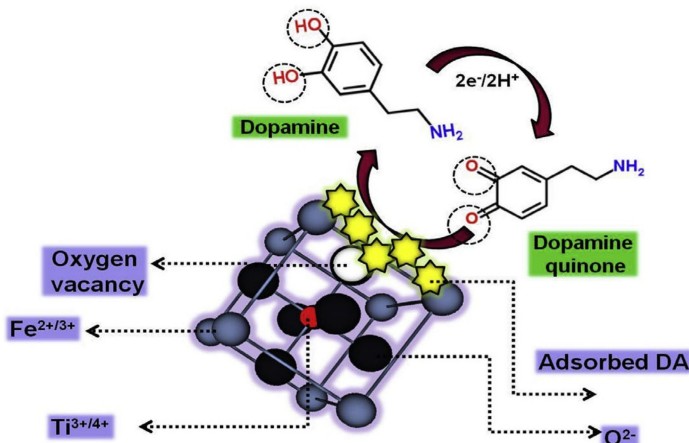

**Figure 5.** Electrooxidation mechanism of DA on $FeTiO_3$ [52]. Reproduced with the permission from Elsevier.

With La-based perovskites, the B-site ions are, as for hydrogen peroxide and glucose detection, Fe, Co, Mn, or Ni. In [49], the electrochemical detection of DA is performed with $LaFeCO_3$, $LaCoO_3$, and $LaNiO_3$ for the comparison of the influence of the B-site cations. The lower detection limit of 9 nM is obtained with $LaNiO_3$. This result was explained by the fact that the energy of the 3d electron present in $Ni^{3+}$ is higher than those in $Co^{3+}$ and $Fe^{3+}$. Then, the energy of the 3d electron may become higher than that of the orbital energy (LUMO) of the analyte in the solution. Thermodynamically favorable energy transfer is then possible towards the $LaNiO_3$-modified electrode, better than towards the other perovskite-modified electrodes. Among other lanthanide A-site ions, it was shown that between neodyme and samarium, Nd-based perovskite ($NdFeO_3$) presents the lower detection limit of 270 nM [56]. Other A-site ions are alkali ions such as sodium [53,55], cesium [59], alkaline-earth ions such as strontium [50,54,58], iron ions [52], or zinc ions [57]. Associated B-site ions, involved in one of the pathways of oxidation of DA to DA hydroquinone (Figure 5), are palladium [50], titanium [52,54], niobium [53,59], iron [55], tin [57], and lead [59]. The lower detection limits (1–3 nM) are obtained with $FeTiO_3$ [53], $\beta$-$NaFeO_2$ [55], $ZnSnO_3$ [57], and $CsPbBr_3$ [59].

Apart from the sensitivity of detection, the simultaneous detection of DA in the presence of other bioactive molecules is another concern. A simultaneous detection of two neurotransmitters, DA and serotonin, in the presence of acetaminophen and tyrosine is obtained with $LaNiO_3$ (Figure 6) [49].

Inadequate levels of DA in human blood sera leads to neurological desorders such as Parkinson's disease, while abnormal levels of UA, xanthine, and hypoxanthine result in gout, pneumonia, and others. Therefore, the simultaneous detection of these biomarkers is of high interest. It has been obtained using electrocatalysis with $\beta$-$NaFeO_2$ perovskite (Figure 7) [55].

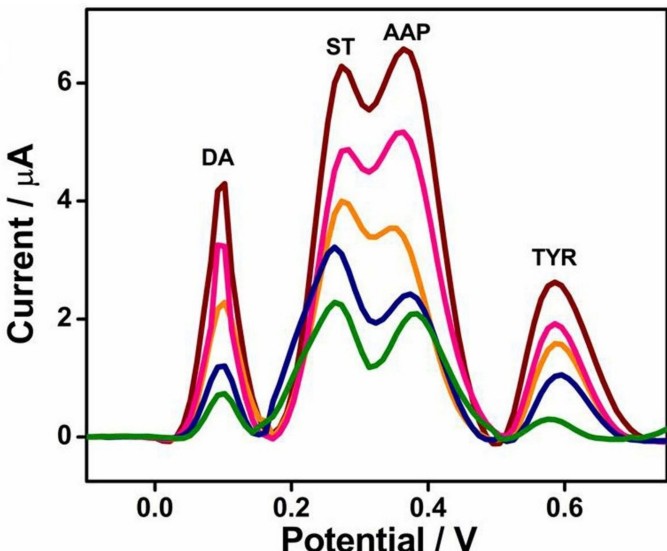

**Figure 6.** Interference measurement of LaNiO$_3$/carbon paste electrode (CPE) sensor using diffential pulse voltammetry (DPV) technique in 0.1 M PBS of pH 7 containing varying concentrations of DA (from green to garnet curve 0.005 to 0.045 mM), serotonine (from green to garnet curve 0.03 to 0.17 mM), acetaminophen (from green to garnet curve 0.03 to 0.09 mM), and tyrosine (from green to garnet curve 0.0008 to 0.04 mM) [49]. Reproduced with the permission from Elsevier.

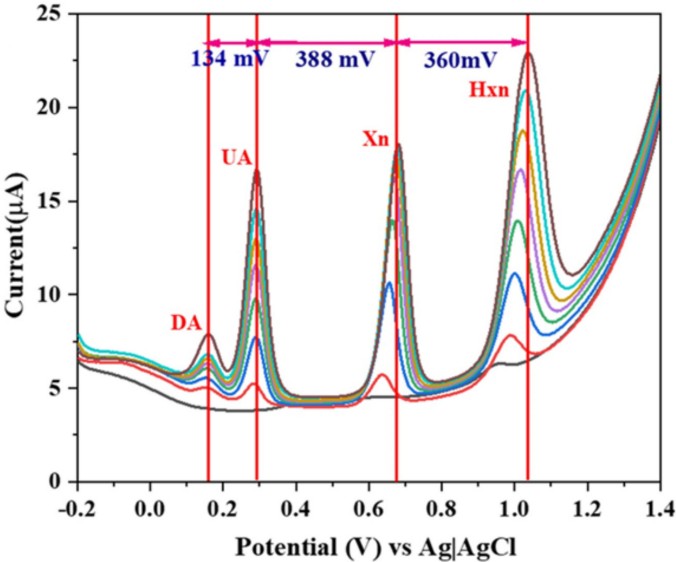

**Figure 7.** DPV response of a β-NaFeO$_2$-modified carbon paste electrode at different concentrations of DA (from red to black curve 10 nM to 70 nM), UA (from red to black curve 5 μM to 200 μM), Xn (from red to black curve 5 μM to 200 μM), and Hxn (from red to black curve 5 μM to 200 μM) [55]. Reproduced with the permission from Elsevier.

## 6. Comparison of Perovskite-Based Electrochemical Sensors with Ni and Co Oxide-Based Electrochemical Sensors

Nickel oxide nanomaterials and cobalt oxide nanomaterials are also used due to their electrocatalytic properties for the detection of electroactive molecules as hydrogen peroxide, glucose, and DA. The works published in the five last years, where the oxide nanomaterials were not associated with other nanomaterials such as graphene, carbon nanotubes, or metallic nanoparticles, are presented in Table 4.

**Table 4.** Nickel oxide and cobalt oxide-based electrochemical sensors for the detection of bioactive molecules.

| Target | Oxide Preparation Electrode | Sensitivity | Linear Range | LOD | Lifetime | Applications | References |
|---|---|---|---|---|---|---|---|
| **NiO nanomaterials** | | | | | | | |
| Glucose | Hydrothermal synthetic approach NiO hollow cages on a Nafion/GCE | 2476.4 $\mu$A/mM/cm$^2$ | 0.1–5.0 mM | 0.1 $\mu$M | 8 weeks | Human serum | [60] |
| Glucose | NiO nanopetals on FTO/glass | 3.9 $\mu$A/$\mu$M/cm$^2$ | 100 $\mu$M–1.2 mM | 1 $\mu$M | | | [61] |
| Glucose | Sol-gel hydrothermal route NiO on ITO | 24 $\mu$A/mM/cm$^2$ | 0.01–83 mM | 8.1 $\mu$M | | | [62] |
| Glucose | Ultrasound-assisted anodization of nickel foils | 206.9 $\mu$A/mM/cm$^2$ | 0.1–10.0 mM | 1.16 $\mu$M | 30 days | | [63] |
| Glucose | Hydrothermal synthetic approach NiO on a gold electrode | 1618.4 $\mu$A/mM/cm$^2$ | 0.25–3.75 mM | 2.5 $\mu$M | | | [64] |
| DA | CTAB-NiO prepared by co-precipitation CPE | | 1–800 $\mu$M | 0.68 $\mu$M | | Human blood serum | [65] |
| DA | Hydrothermal synthetic approach NiO/ITO | 0.064 $\mu$A/$\mu$M | 0.5–5 $\mu$M | 85 nM | | Dopamine release from PC12 cells | [66] |
| DA | Electrodeposited nanoNiOx on a GCE | 0.329 $\mu$A/$\mu$M/cm$^2$ | 80.0–800 $\mu$M | 0.69 $\mu$M | | Vitamin C | [67] |
| **Co$_3$O$_4$ nanomaterials** | | | | | | | |
| H$_2$O$_2$ | 3D porous Co$_3$O$_4$ on a Nafion/GCE | 389.7 $\mu$A/mM/cm$^2$ | 0.4–200 $\mu$M | 0.24 $\mu$M | | | [68] |
| Glucose | Hydrothermal growth of Co$_3$O$_4$ nanowires on a Nafion/GCE | 300.8 $\mu$A/mM/cm$^2$ | 5–570 $\mu$M | 5 $\mu$M | One month | Human serum | [69] |
| Glucose | Hydrothermal growth of Co$_3$O$_4$ nanodiss on a Nafion/GCE | 27.33 $\mu$A/mM/cm$^2$ | 0.5–5.0 mM | 0.8 $\mu$M | | Blood serum samples | [70] |
| Glucose | 3D porous Co$_3$O$_4$ on a Nafion/GCE | 471.5 $\mu$A/mM/cm$^2$ | 1 $\mu$M–12.5 mM | 0.1 $\mu$M | 60 days | Human serum samples | [68] |

Hydrogen peroxide is detected with a 3D porous Co3O4/Nafion-modified GCE at 0.31 V/SCE, in a 0.1 M NaOH solution, and a detection limit of 0.24 µM is obtained [68]. A detection limit of 1 nM is obtained with a $LaNi_{0.6}Co_{0.4}O_3$ carbon paste electrode [28], and a detection limit of 2 nM is obtained with a $Co_{0.4}Fe_{0.6}LaO_3$ carbon paste electrode [29], with the voltammetric signal being obtained at 0.55 V/SCE in 0.1 M NaOH. A detection limit of 23 nM is obtained with a $LaNi_{0.5}Ti_{0.5}O_3/CoFe_2O_3$-modified GCE [30], and a detection limit of 35 nM is obtained with a $LaNiO_3$/Nafion-modified GCE [34], with the voltammetric signal being obtained at 0.6 V/SCE [30] and at $-0.5$ V/SCE (cathodic peak) [34] in 0.1 M NaOH. When the experimental conditions are similar, the detection limit for hydrogen peroxide obtained with lanthanum-based perovskite is lower than that obtained with cobalt oxide nanomaterials.

When glucose is detected with NiO-modified electrodes, the obtained detection limits are between 0.1 µM and 8.1 µM [60–64], with glucose being detected between 0.48 V/SCE and 0.58 V/SCE, in $10^{-4}$ M–0.5 M NaOH. When glucose is detected with $Co_3O_4$-modified electrodes, the obtained detection limits are between 0.1 µM to 5 µM [68–70], with glucose being detected between 0.47 V/SCE and 0.6 V/SCE, in 0.1–0.3 M NaOH. With a $Sr_2Pd_{0.7}Au_{0.3}O_3$-modified graphite electrode, a detection limit of 2.11 nM is obtained [38], and with a $LaNi_{0.6}Co_{0.4}O_3$ carbon paste electrode, a detection limit of 8.0 nM is obtained [28], with glucose being detected in a 0.1 M NaOH solution at $-76$ mV/SCE (with $Sr_2Pd_{0.7}Au_{0.3}O_3$) and at 0.55 V/SCE (with $LaNi_{0.6}Co_{0.4}O_3$). The experimental conditions for glucose detection in the presence of lanthanum-based perovskites are very close to those for glucose detection in the presence of oxide-modified electrodes, but the obtained detection limit is lower. In the presence of Pd-Au-B site strontium perovskite, the detection conditions are quite different (very low oxidation potential), and the detection limit is quite lower [38].

When DA is detected with NiO-modified electrodes, the obtained detection limits are between 85 nM and 690 nM [65–67], with DA being detected at 0.2 V/SCE, at pH between 6 and 7.4. When DA is detected with a $LaNiO_3$ carbon paste electrode, a detection limit of 9 nM is obtained [49], with the detection being obtained at very similar conditions at 0.1 V/SCE and pH 7. When DA is detected with a $FeTiO_3$-modified GCE, a detection limit of 1.3 nM is obtained [52], with the detection being obtained at very similar conditions at 0.15 V/SCE in PBS. Besides the detection limit, another point is the simultaneous detection of DA with possible interfering molecules such as AA and UA. With a $LaFeO_3$-modified GCE, the detection limit of DA is 10 nM, and the positions of the different anodic peaks are 0.25 V/SCE, $-0.05$ V/SCE, and 0.38 V/SCE for DA, AA, and UA, respectively. With NiO-modified electrodes, the anodic peak of DA is at 0.21 V/SCE, and the anodic peaks of AA and UA are at 0.02 V/SCE and at 0.32 V/SCE, respectively. When comparing the DA–AA peak distance, it comes that with a $LaFeO_3$-modified GCE, it is 0.30 V/SCE and with a NiO-modified electrode, it is between 0.19 V/SCE and 0.23 V/SCE, which is lower. When comparing the DA–UA peak distance, it comes that with a $LaFeO_3$-modified GCE, it is 0.13 V/SCE and with a NiO-modified electrode, it is 0.11 V/SCE. It appears that with a $LaFeO_3$-modified GCE, the DA anodic peak is more separated from the interfering peaks.

All these examples show that with lanthanum-based perovskites, with Ni or Co B-site, lower detection limits compared to with nickel and cobalt oxides are obtained and much lower detection limits could be obtained with, for instance, Sr-Pd and Fe-Ti-based perovskites, without any other nanomaterials. Moreover, with $LaFeO_3$, the distance between the anodic peaks of the interfering compounds can be larger.

## 7. Conclusions and Perspectives

A lot of $ABO_3$ perovskites are used for their electrocatalytic properties applied for the sensitive electrochemical detection of bioactive molecules: hydrogen peroxide, glucose, and DA. La-based perovskites, containing Ni and Co ions as B-site, present a higher sensitivity of detection, with the detection limit being in the range of 1–10 nM. The proposed mechanism for the electrooxidation of the bioactive molecules takes place in two pathways

via the $B^{3+}/B^{4+}$ redox couple and via the LOM-OER. The experimental conditions for the hydrogen peroxide, glucose, and DA detection in the presence of lanthanum-based perovskites are very close to those for their detection in the presence of oxide-modified electrodes, but the obtained detection limit is lower due to the involved complex mechanism of detection.

The main trends for improving the analytical performance are the nanostructuration of perovskite materials (nanocrystals, nanoneedles, etc.) for increasing their specific surface areas and the association of perovskites with conductive nanomaterials such as carbon nanotubes, graphene, and metallic nanomaterials. Some perovskites are commercially available; for instance, $CsPbBr_3$ is in form of quantum dots or others are in form of powders or can be used for sputtering. They could also be easily used due to their electrocatalytic properties.

The applications of perovskite-modified electrodes are numerous, due to the large range of detectable molecules and the concerned biomedical, environmental, and agrifood fields. For limiting the environmental impact of perovskite nanomaterials and due to the scarcity and the toxicity of rare earth elements, it is necessary to recycle these elements as well as heavy metals (Ni, Co, and Pb), using solid-state extraction, for instance graphite/magnetite nanocomposites.

**Author Contributions:** Conceptualization, A.Z. and I.B.; resources, A.E.; writing—original draft preparation, I.B.; writing—review and editing, N.J.-R. All authors have read and agreed to the published version of the manuscript.

**Funding:** This research was funded by CAMPUS FRANCE, through PHC Maghreb #39382RE.

**Institutional Review Board Statement:** Not applicable.

**Informed Consent Statement:** Not applicable.

**Acknowledgments:** I.Boubezari thanks the government of Algeria for her work-study grant.

**Conflicts of Interest:** The authors declare no conflict of interest.

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
