# Peer review of "Sensitive Electrochemical Detection of Bioactive Molecules (Hydrogen Peroxide, Glucose, Dopamine) with Perovskites-Based Sensors"

_chemosensors, doi:10.3390/chemosensors9100289_

Round 1

Reviewer 1 Report

Perovskites have rapidly become one of the hot topics in recent research, particularly in the photovoltaic energy era. The authors have presented an interesting subject in the field of bioactive molecule sensing using perovskites-based electrochemical sensors. However, it seems that authors have “listed” papers in this field, but the review is prepared based on recent literature. A broader commentary on listed papers is needed, especially since the tables are the greater part of the paper in relation to the written text. The text is inconsistent and the concept of showing the novelty of the paper is unclear. The authors should decide which element of the text should be emphasized and which one should be omitted. This is particularly evident in the introduction section. The transitions should be used to joining of consecutive paragraphs to make them easier to read by providing connections between them.

Some of the suggested corrections are listed below:

  1. Please consider specifying bioactive molecules in the title.
  2. I suggest using the first part of the conclusions section as the beginning of the introduction section.
  3. APAP is a trademark or is used as the abbreviation of N-acetyl-para-aminophenol.
  4. Section 6 describing drugs detection does not match the text or should be consistently described.
  5. The tables are too long they should be divided into smaller ones (maybe by the substances).

The presented paper requires major revision before any other consideration.

Author Response

Reviewer #1

Perovskites have rapidly become one of the hot topics in recent research, particularly in the photovoltaic energy era. The authors have presented an interesting subject in the field of bioactive molecule sensing using perovskites-based electrochemical sensors. However, it seems that authors have “listed” papers in this field, but the review is prepared based on recent literature. A broader commentary on listed papers is needed, especially since the tables are the greater part of the paper in relation to the written text. The text is inconsistent and the concept of showing the novelty of the paper is unclear. The authors should decide which element of the text should be emphasized and which one should be omitted. This is particularly evident in the introduction section. The transitions should be used to joining of consecutive paragraphs to make them easier to read by providing connections between them.

The introduction was totally revised in order to introduce the novelty of the review paper.

From line 24 to line 44

For the integration in point-of-care testing (POCT) systems [1], such as the commercial kits available for monitoring patients’ glucose level (i. e. Accu-Chek® from Roche Diabetes Care Company) [2], there is an urgent need for precise, sensitive, portable, and cost-effective technologies for the detection of bioactive molecules. Hydrogen peroxide is a cancer biomarker because, in comparison to normal cells, cancer cells are characterized by an increased H2O2 production rate and an impaired redox balance, thereby affecting the antitumoral immune response [3]. Diabetes mellitus is due to an abnormal level of glucose in the blood and this level should be frequently monitored [4] . Dopamine is one of the most important catecholamines, present in the human central nervous system. Its depletion that should be monitored, leads to neurodegenerative diseases such as Parkinson’s disease [5].

Electrochemical sensors are good candidates for their integration in POCT systems [6], due to their easy miniaturization, the low-cost instrumentation that could be interfaced with smart-phone. All these bioactive molecules can be sensitively detected by enzymatic electrochemical sensors [7-9]. The main drawback of these enzymatic sensors is the stability of the enzyme, its activity being modified by the immobilization procedure, by the pH value or by the presence of inhibiting agents. The use of non-enzymatic electrochemical sensors should be highly required. During the last decade, perovskite nanomaterials have shown electrocatalytic properties versus bioactive molecules that could be easily sensitively and selectively detected using perovskite-based electrodes, which present the advantage of stability of the sensor.

And from line 65 to line 74

Several recent review papers were devoted to the sensory applications of perovskite   based on their electrical and of optical properties [11-17]. This review paper is based an exhaustive list of perovskite-based electrochemical sensors for the enzyme-free detection of hydrogen peroxide, of glucose, and of dopamine. In each case the involved mechanism is described and the analytical performance of the obtained sensor is presented. The perovskite formula leading to the lowest detection limit is highlighted, as well as the one leading to the most selective detection. The improvement brought by the association with other nanomaterials is also shown. The analytical performance of the perovskite-based sensors and those of the nickel oxide and cobalt oxide-based sensors are compared. The main trends and future challenges are discussed.

Some of the suggested corrections are listed below:

  1. Please consider specifying bioactive molecules in the title.

The targeted bioactive molecules were cited in the title and limited to: hydrogen peroxide, glucose and dopamine

  1. I suggest using the first part of the conclusions section as the beginning of the introduction section.

The first part of the conclusion was used in the introduction (lines 41-44).

  1. APAP is a trademark or is used as the abbreviation of N-acetyl-para-aminophenol.

This point was introduced in the text (line 199)

  1. Section 6 describing drugs detection does not match the text or should be consistently described.

The targeted bioactive molecules were limited to: hydrogen peroxide, glucose and dopamine. Section 6 decribing drug detection was canceled

  1. The tables are too long they should be divided into smaller ones (maybe by the substances).

The tables were limited to the targeted bioactive molecules and the table remated to drug detection was canceled

The presented paper requires major revision before any other consideration.

Reviewer 2 Report

The Introduction part should be rearranged – first a general „problem“, than description an possibilities in detection of bioactive molecules, then motivation and focus on perovskites-based electrochemical sensors for the detection of bioactive molecules. This backbone would offer an easy to follow for  for a reader. Current Introduction is confusing and a bit messy, since it is not fluid and easy to follow especially in the first half before starting with the perovskites-based electrochemical sensors. Also, only few bioactive molecules are mentioned, which does not make any sense. It depend on the Authors what they would like to stress out about target bioactive molecules, maybe a bioactive molecules type, role and concentration rearrange would be enough...

When Authors stared with „This review article is devoted to the state-of-the art of perovskite-based electrochemical sensors for the detection of molecules of biomedical interest“ it is confusing to mentione it again in „The objective of this review paper is to present an exhaustive list of perovskite-based electrochemical sensors for the enzyme-free detection of hydrogen peroxide….“ Please focus. Introduction part should be informative intro to the rest of the review manuscript.

Also make it clear why this review article is different from the existing articles on the same topic!

Chapter: 2. Methods for synthesis and characterization of perovskites

should be focused on existing methods and a advise to include subtitles for each method to make it more clear for the reader. Please unify the amount of info for each method.

Chapter 5. should be splitted in two separate chapters, A) Perovskite-based electrochemical sensors for the detection of dopamine  AND B) Perovskite-based electrochemical sensors for the detection of other bioactive molecules

Chapter Conclusion should give a brief conclusions and comments on the article.

Author Response

Reviewer 2

The Introduction part should be rearranged – first a general „problem“, than description an possibilities in detection of bioactive molecules, then motivation and focus on perovskites-based electrochemical sensors for the detection of bioactive molecules. This backbone would offer an easy to follow for  for a reader. Current Introduction is confusing and a bit messy, since it is not fluid and easy to follow especially in the first half before starting with the perovskites-based electrochemical sensors. Also, only few bioactive molecules are mentioned, which does not make any sense. It depend on the Authors what they would like to stress out about target bioactive molecules, maybe a bioactive molecules type, role and concentration rearrange would be enough...

When Authors stared with „This review article is devoted to the state-of-the art of perovskite-based electrochemical sensors for the detection of molecules of biomedical interest“ it is confusing to mentione it again in „The objective of this review paper is to present an exhaustive list of perovskite-based electrochemical sensors for the enzyme-free detection of hydrogen peroxide….“ Please focus. Introduction part should be informative intro to the rest of the review manuscript.

Also make it clear why this review article is different from the existing articles on the same topic!

The introduction was totally revised in order to introduce the novelty of the review paper.

From line 24 to line 44

For the integration in point-of-care testing (POCT) systems [1], such as the commercial kits available for monitoring patients’ glucose level (i. e. Accu-Chek® from Roche Diabetes Care Company) [2], there is an urgent need for precise, sensitive, portable, and cost-effective technologies for the detection of bioactive molecules. Hydrogen peroxide is a cancer biomarker because, in comparison to normal cells, cancer cells are characterized by an increased H2O2 production rate and an impaired redox balance, thereby affecting the antitumoral immune response [3]. Diabetes mellitus is due to an abnormal level of glucose in the blood and this level should be frequently monitored [4] . Dopamine is one of the most important catecholamines, present in the human central nervous system. Its depletion that should be monitored, leads to neurodegenerative diseases such as Parkinson’s disease [5].

Electrochemical sensors are good candidates for their integration in POCT systems [6], due to their easy miniaturization, the low-cost instrumentation that could be interfaced with smart-phone. All these bioactive molecules can be sensitively detected by enzymatic electrochemical sensors [7-9]. The main drawback of these enzymatic sensors is the stability of the enzyme, its activity being modified by the immobilization procedure, by the pH value or by the presence of inhibiting agents. The use of non-enzymatic electrochemical sensors should be highly required. During the last decade, perovskite nanomaterials have shown electrocatalytic properties versus bioactive molecules that could be easily sensitively and selectively detected using perovskite-based electrodes, which present the advantage of stability of the sensor.

And from line 65 to line 74

Several recent review papers were devoted to the sensory applications of perovskite   based on their electrical and of optical properties [11-17]. This review paper is based an exhaustive list of perovskite-based electrochemical sensors for the enzyme-free detection of hydrogen peroxide, of glucose, and of dopamine. In each case the involved mechanism is described and the analytical performance of the obtained sensor is presented. The perovskite formula leading to the lowest detection limit is highlighted, as well as the one leading to the most selective detection. The improvement brought by the association with other nanomaterials is also shown. The analytical performance of the perovskite-based sensors and those of the nickel oxide and cobalt oxide-based sensors are compared. The main trends and future challenges are discussed.

Chapter: 2. Methods for synthesis and characterization of perovskites

should be focused on existing methods and a advise to include subtitles for each method to make it more clear for the reader. Please unify the amount of info for each method.

 Chapter 2 was modified accordingly

  1. Methods for synthesis and characterization of perovskites

Sol-gel synthesis

The sol-gel synthesis of the precursors is one of the procedures used for synthesis of the perovskites [18]. It is based on the Pechini method that involves two chemical reactions: nitrate salts of the metal ions being mixed according to the stoichiometry, the complexation of metal ions occurs with the addition of citric acid and polyesterification of the complexes was obtained with the addition of ethylene glycol [19]. A viscous solution is obtained after heating to 130 °C in an argon flow under stirring for about 10 h. The temperature is then raised to 150-200 °C to obtain a foamy dried mass which has to be ground in a mortar. This precursor is calcined at different temperatures of 600, 850 and 1000 °C in a muffle furnace depending of the final targeted product.

Microwave irradiation process

Microwave irradiation process shows many advantages like rapid reaction velocity, uniform heating and clean and energy efficient. The conditions used for microwave preparation are 2.45 GHz and with a maximum output power not less than 1 kW. Dielectric materials can absorb the microwave energy and transform it into heat energy directly through the polarization and dielectric loss in the interior of materials. Single phase manganese-based perovskites were simply obtained from nitric solutions by a denitration process under microwave irradiation [20].

Coprecipitation process

In the coprecipitation process [18], different types of precursors are employed: oxides, alkoxides, inorganic salts and nitrates. The processing parameters (pH, coprecipitation rate, washing, drying and temperature of synthesis) are controlled which results in homogeneous and weakly agglomerated nanopowders. They can be sintered at temperatures as low as 1250 °C and for short times, 1-2 h, for the obtaining of perovskite of near theoretical density.

Solid state synthesis technique

LaFeO3 nanoparticles were prepared through solid state synthesis technique using mechanical ball milling [21]. Stoichiometric amounts of La2O3 and Fe2O3 metal oxide precursors (molar ratio 1:1) were transferred to a planetary ball mill. Wet milling was carried out for 20 h (with toluene as process control agent). Based on thermal analysis, the ball milled powder was calcined at 900 °C for 2 h and then ground into fine powders in an agate mortar and pestle.

Other synthesis techniques

LaNiO3 electrospun nanofibers were prepared by mixing metal salts with PVP followed by sequential calcinations [22]. Thin films of La0.5Sr0.5CoO3-d were obtained by pulsed laser deposition (PLD) technique [23].

Characterization methods

The different phases of the prepared perovskites can be differentiated using X-ray powder diffraction, XRD. In addition, the structure of the perovskite can be characterized using single-crystal XRD analysis. Thermal analysis techniques like TGA, DTA and DSC can be used to test the thermal stability of the prepared perovskites. The different morphological characteristics of the prepared perovskites can be studied using scanning (SEM) and transmission (TEM) electron microscopies. Also, surface area measurement of the prepared perovskites can be carried out using BET. Raman spectroscopy allows the determination of vibration modes in relation with molecular dynamics. In addition, surface chemical groups of the prepared perovskites can be identified using FTIR “Fourier Transform infrared spectroscopy” and XPS “X-ray photo-electron spectroscopy”. The frequency dependent conductivity spectra are determined by using impedance spectroscopy.

Chapter 5. should be splitted in two separate chapters, A) Perovskite-based electrochemical sensors for the detection of dopamine  AND B) Perovskite-based electrochemical sensors for the detection of other bioactive molecules

The targeted bioactive molecules were cited in the title and limited to: hydrogen peroxide, glucose and dopamine. Then the part corresponding to the other bioactive molecules was canceled.

Chapter Conclusion should give a brief conclusions and comments on the article.

Conclusion was modified accordingly.

A lot of ABO3 perovskite were used for their electrocatalytic properties applied for the sensitive electrochemical detection of bioactive molecules: hydrogen peroxide, glucose and dopamine. La-based perovskites, containing Ni and Co ions as B-site, present the higher sensitivity of detection, the detection limit being in the range of 1-10 nM. The proposed mechanism for the electrooxidation of the bioactive molecules takes place in two pathways via the B3+/B4+ redox couple and via the lattice-oxygen mediated oxygen evolution reaction (LOM-OER). The experimental conditions for hydrogen peroxide, glucose and dopamine detection in presence of lanthanum-based perovskite are very close to those for their detection in presence of oxide-modified electrodes but the obtained detection limit is lower, due to the involved complex mechanism of detection.

The main trends for improving the analytical performance are the nanostructuration of perovskite materials (nanocrystals, nanoneedles…) for increasing its specific surface area and the association of perovskite with conductive nanomaterials such as carbon nanotubes, graphene and metallic nanomaterials. Some perovskites are commercially available, CsPbBr3 as quantum dots or others as powders or as targets for sputtering; they could also be easily used for their electrocatalytic properties.

The applications of perovskite-modified electrodes are numerous, due to the large range of detectable molecules and the concerned fields: biomedical, environmental and agrifood. For limiting the environmental impact of perovskite nanomaterials and due to the scarcity and the toxicity of rare earth elements, it is necessary to recycle these elements as well as the heavy metals (Ni, Co, Pb), using solid-state extraction, for instance graphite/magnetite nanocomposites [71].

Round 2

Reviewer 1 Report

Thank you, almost all my concerns have been addressed.

In the introduction section, I suggest following the style presented below:
....bioactive molecules. In this group of chemical compounds (Among them is) is hydrogen peroxide, a cancer biomarker ... an immune response [3]. Diabetes mellitus is due to an abnormal level of glucose in the blood and this level should be frequently monitored [4]. Another target molecule is dopamine, one of the most important catecholamines, ...

Author Response

Reviewer #1

Thank you, almost all my concerns have been addressed.

In the introduction section, I suggest following the style presented below:
....bioactive molecules. In this group of chemical compounds (Among them is) is hydrogen peroxide, a cancer biomarker ... an immune response [3]. Diabetes mellitus is due to an abnormal level of glucose in the blood and this level should be frequently monitored [4]. Another target molecule is dopamine, one of the most important catecholamines, ...

The sentence was modified accordingly

Reviewer 2 Report

The Authors have done considerable effort to modify the manuscript. All changes have been done according to comments.

My only comment is: please remove the reference citation from the Conclusion!

Author Response

Reviewer #2

The Authors have done considerable effort to modify the manuscript. All changes have been done according to comments.

My only comment is: please remove the reference citation from the Conclusion!

The reference was canceled

This manuscript is a resubmission of an earlier submission. The following is a list of the peer review reports and author responses from that submission.

Round 1

Reviewer 1 Report

The authors have presented an interesting subject with an interest in the field of clinical research. The electrochemical sensors have found many valuable applications for different analytical purposes. The authors have compiled recent reports, mainly focused on the modification of the sensor by perovskite nanomaterial. However, there are many reviews about electrochemical sensors particularly perovskite-modified electrodes, and the recent related reviews should be cited in the manuscript. In addition, the authors should classify the novelty of this paper in the introduction section. I have found many reviews in this field, so it seems that the idea is not new, and this paper does not impressive deals with this subject. Nevertheless, the article has scientific soundness and introduce significant merit and is a good compilation of recent applications of this kind of sensors. Overall, some parts of the manuscript are hard to read and contain logical, stylistic, punctuation, typographic errors, whereas some of the sentences are either laconic or unnecessary. It seems that the title is too general do not correspond with the content and the idea of ​​the paper. The authors should make a scientific comment about the work rather than only listing references. Frequently, the next section does not follow the previous one, sometimes it is possible to get lost in the manuscript text. There is a need for scientific storytelling to attract the attention of potential readers but not in the case of this article.

In my opinion, the manuscript is unsuitable for publication in its current form and requires major revision before any other consideration.

Author Response

Manuscript ID: chemosensors-1260850

Type of manuscript: Review

Title: Perovskites for the electrochemical detection of bioactive molecules: a review

Reviewer #1

The authors thank the reviewer for valuable comments that will improve the quality of the manuscript.

The authors have presented an interesting subject with an interest in the field of clinical research. The electrochemical sensors have found many valuable applications for different analytical purposes. The authors have compiled recent reports, mainly focused on the modification of the sensor by perovskite nanomaterial. However, there are many reviews about electrochemical sensors particularly perovskite-modified electrodes, and the recent related reviews should be cited in the manuscript.

The review papers including perovskite-modified electrodes (Refs 8 to 13) are cited in the introduction (lines 72-81)

In addition, the authors should classify the novelty of this paper in the introduction section. I have found many reviews in this field, so it seems that the idea is not new, and this paper does not impressive deals with this subject.

The objective of the present review paper is described in the introduction (lines 81-88)

The objective of this review paper is to present an exhaustive list of perovskite-based electrochemical sensors for the enzyme-free detection of hydrogen peroxide, of glucose, of dopamine and other bioactive molecules and drugs. In each case the involved mechanism is described and the analytical performance of the obtained sensor is presented. The perovskite formula leading to the lowest detection limit is highlighted, as well as the one leading to the most selective detection. The improvement brought by the association with other nanomaterials is also shown.

Nevertheless, the article has scientific soundness and introduce significant merit and is a good compilation of recent applications of this kind of sensors. Overall, some parts of the manuscript are hard to read and contain logical, stylistic, punctuation, typographic errors, whereas some of the sentences are either laconic or unnecessary.

It seems that the title is too general do not correspond with the content and the idea of ​​the paper.

The title was modified as follows:

“Perovskites-based electrochemical sensors for the detection of bioactive molecules: a review”

The authors should make a scientific comment about the work rather than only listing references. Frequently, the next section does not follow the previous one, sometimes it is possible to get lost in the manuscript text. There is a need for scientific storytelling to attract the attention of potential readers but not in the case of this article. In my opinion, the manuscript is unsuitable for publication in its current form and requires major revision before any other consideration.

A scientific comment about the obtained analytical performance in connection with the perovskite composition and structure is presented for each class of target molecule.

Reviewer 2 Report

The review describes the application of perovskites as sensorial material for the determination of biomolecules. I found the review a bit short in terms of the sensors response mechanism. But this doesn't significantly affect the quality of the review. 

Here are some suggestions:

1) Figure 2 and Figure 5

The figures are very similar. I suggest deleting Figure 2.

2) I know it's hard to get permission from publishers to copy pictures. But I suggest the authors add some more figures in the review, mainly about sensor response mechanism.

3) 7. Conclusion and perspectives

The authors can describe what the environmental impact of these materials would be.

Author Response

Manuscript ID: chemosensors-1260850

Type of manuscript: Review

Title: Perovskites for the electrochemical detection of bioactive molecules: a review

Reviewer #2

The authors thank the reviewer for valuable comments that will improve the quality of the manuscript.

The review describes the application of perovskites as sensorial material for the determination of biomolecules. I found the review a bit short in terms of the sensors response mechanism. But this doesn't significantly affect the quality of the review. 

Here are some suggestions:

1) Figure 2 and Figure 5

The figures are very similar. I suggest deleting Figure 2.

Figure 5 was deleted.

2) I know it's hard to get permission from publishers to copy pictures. But I suggest the authors add some more figures in the review, mainly about sensor response mechanism.

A new Fig. 5 was added about response mechanism for dopamine.

3) 7. Conclusion and perspectives

The authors can describe what the environmental impact of these materials would be.

A last sentence was added about the environmental impact of perovskite nanomaterials

“For limiting the environmental impact of perovskite nanomaterials and due to the scarcity and the toxicity of rare earth elements, it is necessary to recycle these elements as well as the heavy metals (Ni, Co, Pb), using solid state extraction, for instance graphite/magnetite nanocomposites [77].”

Reviewer 3 Report

Here are my comments/suggestions: Introduction- This section is very poor. Extensive literature review is required. The authors should cover most of the major works starting from its discovery to the present time with a major focus as electrochemical sensors. A summary table highlighting the sensor, target, sensitivity, detection limit, type of transducer, etc. should be included. For the synthesis part, it would be nice if the authors can show a schematic diagram generalizing how perovskites are usually made. Authors are also required to talk about other types of transducers except electrochemical. For each target molecule, can you please also provide a table summary covering the major parameters? Overall, the paper is very weak as it doesn't necessarily provide much impact to the readers in this field. It lacks scientific soundness and detailed insight.

Author Response

Manuscript ID: chemosensors-1260850

Type of manuscript: Review

Title: Perovskites for the electrochemical detection of bioactive molecules: a review

Reviewer #3

The authors thank the reviewer for valuable comments that will improve the quality of the manuscript.

Here are my comments/suggestions: Introduction- This section is very poor. Extensive literature review is required. The authors should cover most of the major works starting from its discovery to the present time with a major focus as electrochemical sensors. A summary table highlighting the sensor, target, sensitivity, detection limit, type of transducer, etc. should be included. For the synthesis part, it would be nice if the authors can show a schematic diagram generalizing how perovskites are usually made. Authors are also required to talk about other types of transducers except electrochemical.

The objective of this review paper is limited to the obtained analytical performance using perovskite- modified electrodes for the detection of bioactive molecules. This objective is presented in the introduction (lines 81-88).

The objective of this review paper is to present an exhaustive list of perovskite-based electrochemical sensors for the enzyme-free detection of hydrogen peroxide, of glucose, of dopamine and other bioactive molecules and drugs. In each case the involved mechanism is described and the analytical performance of the obtained sensor is presented. The perovskite formula leading to the lowest detection limit is highlighted, as well as the one leading to the most selective detection. The improvement brought by the association with other nanomaterials is also shown.

For each target molecule, can you please also provide a table summary covering the major parameters? Overall, the paper is very weak as it doesn't necessarily provide much impact to the readers in this field. It lacks scientific soundness and detailed insight.

A scientific comment about the obtained analytical performance in connection with the perovskite composition and structure is presented for each class of target molecule.